# Semantic Conditioned Dynamic Modulation for Temporal Sentence Grounding in Videos

**Yitian Yuan**\*
Tsinghua-Berkeley Shenzhen Institute
Tsinghua University
`yyt18@mails.tsinghua.edu.cn`

**Lin Ma**
Tencent AI Lab
`forest.linma@gmail.com`

**Jingwen Wang**
Tencent AI Lab
`jaywongjaywong@gmail.com`

**Wei Liu**
Tencent AI Lab
`wl2223@columbia.edu`

**Wenwu Zhu**
Tsinghua University
`wwzhu@tsinghua.edu.cn`

## Abstract

Temporal sentence grounding in videos aims to detect and localize one target video segment, which semantically corresponds to a given sentence. Existing methods mainly tackle this task via matching and aligning semantics between a sentence and candidate video segments, while neglect the fact that the sentence information plays an important role in temporally correlating and composing the described contents in videos. In this paper, we propose a novel semantic conditioned dynamic modulation (SCDM) mechanism, which relies on the sentence semantics to modulate the temporal convolution operations for better correlating and composing the sentence-related video contents over time. More importantly, the proposed SCDM performs dynamically with respect to the diverse video contents so as to establish a more precise matching relationship between sentence and video, thereby improving the temporal grounding accuracy. Extensive experiments on three public datasets demonstrate that our proposed model outperforms the state-of-the-arts with clear margins, illustrating the ability of SCDM to better associate and localize relevant video contents for temporal sentence grounding. Our code for this paper is available at https://github.com/yytzsy/SCDM .

## 1 Introduction

Detecting or localizing activities in videos [18, 30, 26, 34, 11, 29, 21, 9, 8] is a prominent while fundamental problem for video understanding. As videos often contain intricate activities that cannot be indicated by a predefined list of action classes, a new task, namely temporal sentence grounding in videos (TSG) [14, 10], has recently attracted much research attention [2, 36, 3, 19, 4, 5, 35]. Formally, given an untrimmed video and a natural sentence query, the task aims to identify the start and end timestamps of one specific video segment, which contains activities of interest semantically corresponding to the given sentence query.

Most of existing approaches [10, 14, 19, 4] for the TSG task often sample candidate video segments first, then fuse the sentence and video segment representations together, and thereby evaluate their matching relationships based on the fused features. Lately, some approaches [2, 36] try to directly fuse the sentence information with each video clip, then employ an LSTM or a ConvNet to compose the fused features over time, and thus predict the temporal boundaries of the target video segment. While promising results have been achieved, there are still several problems that need to be concerned.

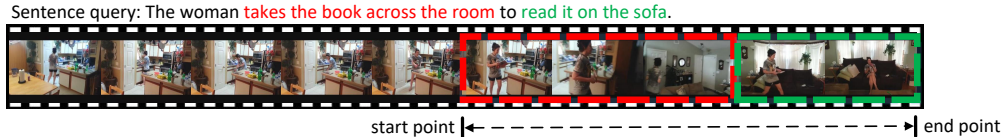

Figure 1: The temporal sentence grounding in videos (TSG) task. Our proposed SCDM relies on the sentence to modulate the temporal convolution operations, which can thereby temporally correlate and compose the various sentence-related activities (highlighted in red and green) for more accurate grounding results.

First, previous methods mainly focus on semantically matching sentences and individual video segments or clips, while neglect the important guiding role of sentences to help correlate and compose video contents over time. For example, the target video sequence shown in Figure 1 mainly expresses two distinct activities "*woman walks cross the room*" and "*woman reads the book on the sofa*". Without referring to the sentence, these two distinct activities are not easy to be associated as one whole event. However, the sentence clearly indicates that "*The woman takes the book across the room to read it on the sofa*". Keeping such a semantic meaning in mind, persons can easily correlate the two activities together and thereby precisely determine the temporal boundaries. Therefore, how to make use of the sentence semantics to guide the composing and correlating of relevant video contents over time is very crucial for the TSG task. Second, activities contained in videos are usually of diverse visual appearances, and present in various temporal scales. Therefore, the sentence guidance for composing and correlating video contents should also be considered in different temporal granularities and dynamically evolve with the diverse visual appearances.

In this paper, we propose a novel semantic conditioned dynamic modulation (SCDM) mechanism, which leverages sentence semantic information to modulate the temporal convolution processes in a hierarchical temporal convolutional network. The SCDM manipulates the temporal feature maps by adjusting the scaling and shifting parameters for feature normalization with referring to the sentence semantics. As such, the temporal convolution process is activated to better associate and compose sentence-related video contents over time. More specifically, such a modulation dynamically evolves when processing different convolutional layers and different locations of feature maps, so as to better align the sentence and video semantics under diverse video contents and various granularities. Coupling SCDM with the temporal convolutional network, our proposed model naturally characterizes the interaction behaviors between sentence and video, leading to a novel and effective architecture for the TSG task.

Our main contributions are summarized as follows. (1) We propose a novel semantic conditioned dynamic modulation (SCDM) mechanism, which dynamically modulates the temporal convolution procedure by referring to the sentence semantic information. In doing so, the sentence-related video contents can be temporally correlated and composed to yield a precise temporal boundary prediction. (2) Coupling the proposed SCDM with the hierarchical temporal convolutional network, our model naturally exploits the complicated semantic interactions between sentence and video in various temporal granularities. (3) We conduct experiments on three public datasets, and verify the effectiveness of the proposed SCDM mechanism as well as its coupled temporal convolution architecture with the superiority over the state-of-the-art methods.

## 2 Related Works

Temporal sentence grounding in videos is a new task introduced recently [10, 14]. Some previous works [10, 14, 19, 33, 4, 12] often adopted a two-stage multimodal matching strategy to solve this problem. They sampled candidate segments from a video first, then integrated the sentence representation with those video segments individually, and thus evaluated their matching relationships through the integrated features. With the above multimodal matching framework, Hendricks *et al.* [14] further introduced temporal position features of video segments into the feature fusion procedure; Gao *et al.* [10] established a location regression network to adjust the temporal position of the candidate segment to the target segment; Liu *et al.* [19] designed a memory attention mechanism to emphasize the visual features mentioned in the sentence; Xu *et al.* [33] and Chen *et al.* [4] proposed to generate query-specific proposals as candidate segments; Ge *et al.* [12] investigated activity concepts from both videos and queries to enhance the temporal sentence grounding.

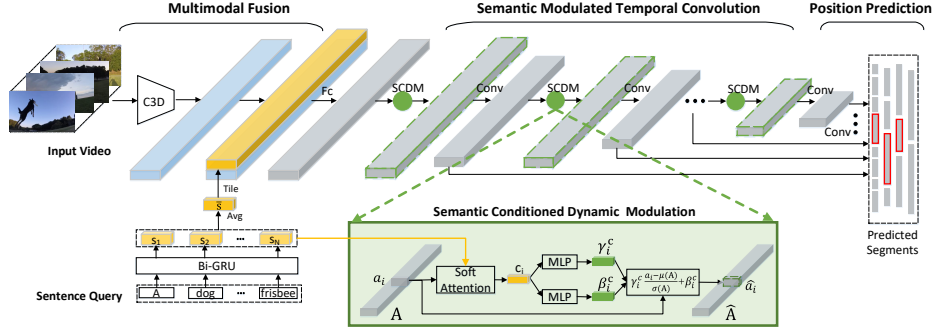

Figure 2: An overview of our proposed model for the TSG task, which consists of three fully-coupled components. The multimodal fusion fuses the entire sentence and each video clip in a fine-grained manner. Based on the fused representation, the semantic modulated temporal convolution correlates sentence-related video contents in the temporal convolution procedure, with the proposed SCDM dynamically modulating temporal feature maps with reference to the sentence. Finally, the position prediction outputs the location offsets and overlap scores of candidate video segments based on the modulated features. Best viewed in color.

Recently, some other works [4, 36] proposed to directly integrate sentence information with each fine-grained video clip unit, and then predicted the temporal boundary of the target segment by gradually merging the fusion feature sequence over time in an end-to-end fashion. Specifically, Chen *et al.* [2] aggregated frame-by-word interactions between video and language through a Match-LSTM [31]. Zhang *et al.* [36] adopted the Graph Convolutional Network (GCN) [16] to model relations among candidate segments produced from a convolutional neural network.

Although promising results have been achieved by existing methods, they all focus on better aligning semantic information between sentence and video, while neglect the fact that sentence information plays an important role in correlating the described activities in videos. Our work firstly introduces the sentence information as a critical prior to compose and correlate video contents over time, subsequently sentence-guided video composing is dynamically performed and evolved in a hierarchical temporal convolution architecture, in order to cover the diverse video contents of various temporal granularities.

## 3 The Proposed Model

Given an untrimmed video $V$ and a sentence query $S$, the TSG task aims to determine the start and end timestamps of one video segment, which semantically corresponds to the given sentence query. In order to perform the temporal grounding, the video is first represented as $\mathbf{V} = \{\mathbf{v}_t\}_{t=1}^{T}$ clip-by-clip, and accordingly the query sentence is represented as $\mathbf{S} = \{\mathbf{s}_n\}_{n=1}^{N}$ word-by-word.

In this paper, we propose one novel model to handle the TSG task, as illustrated in Figure 2. Specifically, the proposed model consists of three components, namely the multimodal fusion, the semantic modulated temporal convolution, and the position prediction. Please note that the three components fully couple together and can therefore be trained in an end-to-end manner.

### 3.1 Multimodal Fusion

The TSG task requires to understand both the sentence and video. As such, in order to correlate their corresponding semantic information, we first let each video clip meet and interact with the entire sentence, which is formulated as:

$$\mathbf{f}_t = \text{ReLU}\left(\mathbf{W}^f\left(\mathbf{v}_t \| \bar{\mathbf{s}}\right) + \mathbf{b}^f\right), \tag{1}$$

where $\mathbf{W}^f$ and $\mathbf{b}^f$ are the learnable parameters. $\bar{\mathbf{s}}$ denotes the global sentence representation, which can be obtained by simply averaging the word-level sentence representation $\mathbf{S}$. With such a multimodal fusion strategy, the yielded representation $\mathbf{F} = \{\mathbf{f}_t\}_{t=1}^{T} \in \mathcal{R}^{T \times d_f}$ captures the interactions between sentence and video clips in a fine-grained manner. The following semantic modulated temporal convolution will gradually correlate and compose such representations together over time, expecting to help produce accurate temporal boundary predictions of various scales.

## 3.2 Semantic Modulated Temporal Convolution

As aforementioned, the sentence-described activities in videos may have various durations and scales. Therefore, the fused multimodal representation $\mathbf{F}$ should be exploited from different temporal scales to comprehensively characterize the temporal diversity of video activities. Inspired by the efficient single-shot object and action detections [20, 18], the temporal convolutional network established via one hierarchical architecture is used to produce multi-scale features to cover the activities of various durations. Moreover, in order to fully exploit the guiding role of the sentence, we propose one novel semantic conditioned dynamic modulation (SCDM) mechanism, which relies on the sentence semantics to modulate the temporal convolution operations for better correlating and composing the sentence-related video contents over time. In the following, we first review the basics of the temporal convolutional network. Afterwards, the proposed SCDM will be described in details.

### 3.2.1 Temporal Convolutional Network

Taking the multimodal fusion representation $\mathbf{F}$ as input, the standard temporal convolution operation in this paper is denoted as $\text{Conv}(\theta_k, \theta_s, d_h)$, where $\theta_k$, $\theta_s$, and $d_h$ indicate the kernel size, stride size, and filter numbers, respectively. Meanwhile, the nonlinear activation, such as ReLU, is then followed with the convolution operation to construct a basic temporal convolutional layer. By setting $\theta_k$ as 3 and $\theta_s$ as 2, respectively, each convolutional layer will halve the temporal dimension of the input feature map and meanwhile expand the receptive field of each feature unit within the map. By stacking multiple layers, a hierarchical temporal convolutional network is constructed, with each feature unit in one specific feature map corresponding to one specific video segment in the original video. For brevity, we denote the output feature map of the $k$-th temporal convolutional layer as $\mathbf{A}_k = \{\mathbf{a}_{k,i}\}_{i=1}^{T_k} \in \mathcal{R}^{T_k \times d_h}$, where $T_k = T_{k-1}/2$ is the temporal dimension, and $\mathbf{a}_{k,i} \in \mathbf{R}^{d_h}$ denotes the $i$-th feature unit at the the $k$-th layer feature map.

### 3.2.2 Semantic Conditioned Dynamic Modulation

Regarding video activity localization, besides the video clip contents, their temporal correlations play an even more important role. For the TSG task, the query sentence, presenting rich semantic indications on such important correlations, provides crucial information to temporally associate and compose the consecutive video contents over time. Based on the above considerations, in this paper, we propose a novel SCDM mechanism, which relies on the sentence semantic information to dynamically modulate the feature composition process in each temporal convolutional layer.

Specifically, as shown in Figure 3(b), given the sentence representation $\mathbf{S} = \{\mathbf{s}_n\}_{n=1}^N$ and one feature map extracted from one specific temporal convolutional layer $\mathbf{A} = \{\mathbf{a}_i\}$ (we omit the layer number here), we attentively summarize the sentence representation to $\mathbf{c}_i$ with respect to each feature unit $\mathbf{a}_i$:

$$\rho_i^n = \text{softmax}\big(\mathbf{w}^\top \tanh\big(\mathbf{W}^s\mathbf{s}_n + \mathbf{W}^a\mathbf{a}_i + \mathbf{b}\big)\big), \qquad \mathbf{c}_i = \sum_{n=1}^N \rho_i^n \mathbf{s}_n, \qquad (2)$$

where $\mathbf{w}$, $\mathbf{W}^s$, $\mathbf{W}^a$, and $\mathbf{b}$ are the learnable parameters. Afterwards, two fully-connected (FC) layers with the $\texttt{tanh}$ activation function are used to generate two modulation vectors $\gamma_i^c \in \mathbf{R}^{d_h}$ and $\beta_i^c \in \mathbf{R}^{d_h}$, respectively:

$$\begin{aligned} \gamma_i^c &= \tanh(\mathbf{W}^\gamma \mathbf{c}_i + \mathbf{b}^\gamma), \\ \beta_i^c &= \tanh(\mathbf{W}^\beta \mathbf{c}_i + \mathbf{b}^\beta), \end{aligned} \qquad (3)$$

where $\mathbf{W}^\gamma$, $\mathbf{b}^\gamma$, $\mathbf{W}^\beta$, and $\mathbf{b}^\beta$ are the learnable parameters. Finally, based on the generated modulation vectors $\gamma_i^c$ and $\beta_i^c$, the feature unit $\mathbf{a}_i$ is modulated as:

$$\hat{\mathbf{a}}_i = \gamma_i^c \cdot \frac{\mathbf{a}_i - \mu(\mathbf{A})}{\sigma(\mathbf{A})} + \beta_i^c. \qquad (4)$$

With the proposed SCDM, the temporal feature maps, yielded during the temporal convolution process, are

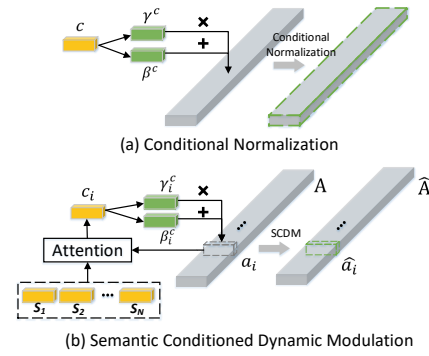

(a) Conditional Normalization

(b) Semantic Conditioned Dynamic Modulation

Figure 3: The comparison between conditional normalization and our proposed semantic conditioned dynamic modulation.

meticulously modulated by scaling and shifting the corresponding normalized features under the sentence guidance. As such, each temporal feature map will absorb the sentence semantic information, and further activate the following temporal convolutional layer to better correlate and compose the sentence-related video contents over time. Coupling the proposed SCDM with each temporal convolutional layer, we thus obtain the novel semantic modulated temporal convolution as shown in the middle part of Figure 2.

**Discussion.** As shown in Figure 3, our proposed SCDM differs from the existing conditional batch/instance normalization [6, 7], where the same $\gamma^c$ and $\beta^c$ are applied within the whole batch/instance. On the contrary, as indicated in Equations (2)-(4), our SCDM dynamically aggregates the meaningful words with referring to different video contents, making the yielded $\gamma^c$ and $\beta^c$ dynamically evolve for different temporal units within each specific feature map. Such a dynamic modulation enables each temporal feature unit to be interacted with each word to collect useful grounding cues along the temporal dimension. Therefore, the sentence-video semantics can be better aligned over time to support more precise boundary predictions. Detailed experimental demonstrations will be given in Section 4.5.

## 3.3 Position Prediction

Similar to [20, 18] for object/action detections, during the prediction, lower and higher temporal convolutional layers are used to localize short and long activities, respectively. As illustrated in Figure 4, regarding a feature map with temporal dimension $T_k$, the basic temporal span for each feature unit within this feature map is $1/T_k$. We impose different scale ratios based on the basic span, and denote them as $r \in R = \{0.25, 0, 5, 0.75, 1.0\}$. As such, for the $i$-th feature unit of the feature map, we can compute the length of the scaled spans within it as $r/T_k$, and the center of these spans is $(i+0.5)/T_k$. For the whole feature map, there are a total number of $T_k \cdot |R|$ scaled spans within it, with each span corresponding to a candidate video segment for grounding.

Then, we impose an additional set of convolution operations on the layer-wise temporal feature maps to predict the target video segment position. Specifically, each candidate segment will be associated with a prediction vector $p = (p^{over}, \triangle c, \triangle w)$, where $p^{over}$ is the predicted overlap score between the candidate and ground-truth segment, and $\triangle c$ and $\triangle w$ are the temporal center and width offsets of the candidate segment relative to the ground-truth. Suppose

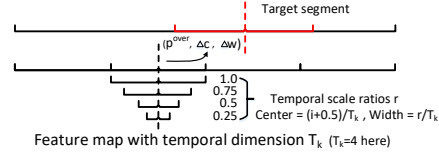

Figure 4: The illustration of temporal scale ratios and offsets.

that the center and width for a candidate segment are $\mu^c$ and $\mu^w$, respectively. Then the center $\phi^c$ and width $\phi^w$ of the corresponding predicted segment are therefore determined by:

$$\phi^c = \mu^c + \alpha^c \cdot \mu^w \cdot \triangle c, \qquad \phi^w = \mu^w \cdot exp(\alpha^w \cdot \triangle w), \tag{5}$$

where $\alpha^c$ and $\alpha^w$ both are used for controlling the effect of location offsets to make location prediction stable, which are set as 0.1 empirically. As such, for a feature map with temporal dimension $T_k$, we can obtain a predicted segment set $\Phi_k = \{(p_j^{over}, \phi_j^c, \phi_j^w)\}_{j=1}^{T_k \cdot |R|}$. The total predicted segment set is therefore denoted as $\boldsymbol{\Phi} = \{\Phi_k\}_{k=1}^K$, where $K$ is the number of temporal feature maps.

## 3.4 Training and Inference

**Training:** Our training sample consists of three elements: an input video, a sentence query, and the ground-truth segment. We treat candidate segments within different temporal feature maps as positive if their tIoUs (temporal Intersection-over-Union) with ground-truth segments are larger than 0.5. Our training objective includes an overlap prediction loss $L_{over}$ and a location prediction loss $L_{loc}$. The $L_{over}$ term is realized as a cross-entropy loss, which is defined as:

$$L_{over} = \sum_{z \in \{pos, neg\}} -\frac{1}{N_z} \sum_i^{N_z} g_i^{over} \log(p_i^{over}) + (1 - g_i^{over}) \log(1 - p_i^{over}), \tag{6}$$

where $g^{over}$ is the ground-truth tIoU between the candidate and target segments, and $p^{over}$ is the predicted overlap score. The $L_{loc}$ term measures the Smooth $L_1$ loss [13] for positive samples:

$$L_{loc} = \frac{1}{N_{pos}} \sum_i^{N_{pos}} SL_1(g_i^c - \phi_i^c) + SL_1(g_i^w - \phi_i^w), \qquad (7)$$

where $g^c$ and $g^w$ are the center and width of the ground-truth segment, respectively.

The two losses are jointly considered for training our proposed model, with $\lambda$ and $\eta$ balancing their contributions:

$$L_{all} = \lambda L_{over} + \eta L_{loc}. \qquad (8)$$

**Inference:** The predicted segment set $\Phi$ of different temporal granularities can be generated in one forward pass. All the predicted segments within $\Phi$ will be ranked and refined with non maximum suppression (NMS) according to the predicted $p^{over}$ scores. Afterwards, the final temporal grounding result is obtained.

## 4 Experiments

### 4.1 Datasets and Evaluation Metrics

We validate the performance of our proposed model on three public datasets for the TSG task: TACoS [24], Charades-STA [10], and AcitivtyNet Captions [17]. The TACoS dataset mainly contains videos depicting human's cooking activities, while Charades-STA and ActivityNet Captions focus on more complicated human activities in daily life.

For fair comparisons, we adopt "R@n, IoU@m" as our evaluation metrics as in previous works [2, 10, 36, 32, 19]. Specifically, "R@n, IoU@m" is defined as the percentage of the testing queries having at least one hitting retrieval (with IoU larger than $m$) in the top-$n$ retrieved segments.

### 4.2 Implementation Details

Following the previous methods, 3D convolutional features (C3D [28] for TACoS and ActivityNet, and I3D [1] for Charades-STA) are extracted to encode videos, with each feature representing a 1-second video clip. According to the video duration statistics, the length of input video clips is set as 1024 for both ActivityNet Captions and TACoS, and 64 for Charades-STA to accommodate the temporal convolution. Longer videos are truncated, and shorter ones are padded with zero vectors. For the design of temporal convolutional layers, 6 layers with {32, 16, 8, 4, 2, 1} temporal dimensions, 6 layers with {512, 256, 128, 64, 32, 16} temporal dimensions, and 8 layers with {512, 256, 128, 64, 32, 16, 8, 4} temporal dimensions are set for Charades-STA, TACoS, and ActivityNet Captions, respectively. All the first temporal feature maps will not be used for location prediction, because the receptive fields of the corresponding feature units are too small and are too rare to contain target activities. To save model memory footprint, the SCDM mechanism is only performed on the following temporal feature maps which directly serve for position prediction. For sentence encoding, we first embed each word in sentences with the Glove [23], and then employ a Bi-directional GRU to encode the word embedding sequence. As such, words in sentences are finally represented with their corresponding GRU hidden states. Hidden dimension of the sentence Bi-directional GRU, dimension of the multimodal fused features $d_f$, and the filter number $d_h$ for temporal convolution operations are all set as 512 in this paper. The trade-off parameters of the two loss terms $\lambda$ and $\eta$ are set as 100 and 10, respectively.

### 4.3 Compared Methods

We compare our proposed model with the following state-of-the-art baseline methods on the TSG task. **CTRL** [10]: Cross-model Temporal Regression Localizer. **ACRN** [19]: Attentive Cross-Model Retrieval Network. **TGN** [2]: Temporal Ground-Net. **MCF** [32]: Multimodal Circulant Fusion. **ACL** [12]: Activity Concepts based Localizer. **SAP** [4]: A two-stage approach based on visual concept mining. **Xu *et al.*** [33]: A two-stage method (proposal generation + proposal rerank) exploiting sentence re-construction. **MAN** [36]: Moment Alignment Network. We use **Ours-SCDM** to refer our temporal convolutional network coupled with the proposed SCDM mechanism.

Table 1: Performance comparisons on the TACoS and Charades-STA datasets (%).

| Method | TACoS | | | | Charades-STA | | | |
|---|---|---|---|---|---|---|---|---|
| | R@1, IoU@0.3 | R@1, IoU@0.5 | R@5, IoU@0.3 | R@5, IoU@0.5 | R@1, IoU@0.5 | R@1, IoU@0.7 | R@5, IoU@0.5 | R@5, IoU@0.7 |
| CTRL (C3D) [10] | 18.32 | 13.30 | 36.69 | 25.42 | 23.63 | 8.89 | 58.92 | 29.52 |
| MCF (C3D) [32] | 18.64 | 12.53 | 37.13 | 24.73 | - | - | - | - |
| ACRN (C3D) [19] | 19.52 | 14.62 | 34.97 | 24.88 | - | - | - | - |
| SAP (VGG) [4] | - | 18.24 | - | 28.11 | 27.42 | 13.36 | 66.37 | 38.15 |
| ACL (C3D) [12] | 24.17 | 20.01 | **42.15** | 30.66 | 30.48 | 12.20 | 64.84 | 35.13 |
| TGN (C3D) [2] | 21.77 | 18.90 | 39.06 | 31.02 | - | - | - | - |
| Xu et al. (C3D) [33] | - | - | - | - | 35.60 | 15.80 | 79.40 | 45.40 |
| MAN (I3D) [36] | - | - | - | - - | 46.53 | 22.72 | **86.23** | 53.72 |
| **Ours-SCDM** (*) | **26.11** | **21.17** | 40.16 | **32.18** | **54.44** | **33.43** | 74.43 | **58.08** |

*: We adopt C3D [28] features to encode videos on the TACoS and ActivityNet Captions datasets, and I3D [1] features on the Charades-STA dataset for fair comparisons. Video features adopted by other compared methods are indicated in brackets. VGG denotes VGG16 [25] features.

Table 2: Performance comparisons on the ActivityNet Captions dataset (%).

| Method | R@1,IoU@0.3 | R@1,IoU@0.5 | R@1,IoU@0.7 | R@5,IoU@0.3 | R@5,IoU@0.5 | R@5,IoU@0.7 |
|---|---|---|---|---|---|---|
| TGN (INP*) [2] | 45.51 | 28.47 | - | 57.32 | 43.33 | - |
| Xu et al. (C3D) [33] | 45.30 | 27.70 | 13.60 | 75.70 | 59.20 | 38.30 |
| **Ours-SCDM** (C3D) | **54.80** | **36.75** | **19.86** | **77.29** | **64.99** | **41.53** |

*: INP denotes Inception-V4 [27] features.

## 4.4 Performance Comparison and Analysis

Table 1 and Table 2 report the performance comparisons between our model and the existing methods on the aforementioned three public datasets. Overall, Ours-SCDM achieves the highest temporal sentence grounding accuracy, demonstrating the superiority of our proposed model. Notably, for localizing complex human activities in Charades-STA and ActivityNet Captions datasets, Ours-SCDM significantly outperforms the state-of-the-art methods with 10.71% and 6.26% absolute improvements in the R@1,IoU@0.7 metrics, respectively. Although Ours-SCDM achieves lower results of R@5,IoU@0.5 on the Charades-STA dataset, it is mainly due to the biased annotations in this dataset. For example, in Charades-STA, the annotated ground-truth segments are 10s on average while the video duration is only 30s on average. Randomly selecting one candidate segment can also achieve competing temporal grounding results. It indicates that the Recall values under higher IoUs are more stable and convincing even considering the dataset biases. The performance improvements under the high IoU threshold demonstrate that Ours-SCDM can generate grounded video segments of more precise boundaries. For TACoS, the cooking activities take place in the same kitchen scene with some slightly varied cooking objects (*e.g.*, chopping board, knife, and bread, as shown in the second example of Figure 5). Thus, it is hard to localize such fine-grained activities. However, our proposed model still achieves the best results, except slight worse performances in R@5,IoU@0.3.

The main reasons for our proposed model outperforming the competing models lie in two folds. First, the sentence information is fully leveraged to modulate the temporal convolution processes, so as to help correlate and compose relevant video contents over time to support the temporal boundary prediction. Second, the modulation procedure dynamically evolves with different video contents in the hierarchical temporal convolution architecture, and therefore characterizes the diverse sentence-video semantic interactions of different granularities.

## 4.5 Ablation Studies

In this section, we perform ablation studies to examine the contributions of our proposed SCDM. Specifically, we re-train our model with the following four settings.

- **Ours-w/o-SCDM**: SCDM is replaced by the plain batch normalization [15].
- **Ours-FC**: Instead of performing SCDM, one FC layer is used to fuse each temporal feature unit with the global sentence representation $\bar{s}$ after each temporal convolutional layer.
- **Ours-MUL**: Instead of performing SCDM, element-wise multiplication between each temporal feature unit and the global sentence representation $\bar{s}$ is performed after each temporal convolutional layer.
- **Ours-SCM**: We use the global sentence representation $\bar{s}$ to produce $\gamma^c$ and $\beta^c$ without dynamically changing these two modulation vectors with respect to different feature units.

Table 3 shows the performance comparisons of our proposed full model Ours-SCDM w.r.t. these ablations on the Charades-STA dataset (please see results on the other datasets in our supplemental material). Without considering SCDM, the performance of the model Ours-w/o-SCDM degenerates dramatically. It indicates that only relying on multimodal fusion to exploit the relationship between video and sentence is not enough for the TSG task. The critical sentence semantics should be intensified to guide the temporal convolution procedure so as to better link the sentence-related video contents over time. However, roughly introducing sentence information in the temporal convolution architecture like Ours-MUL and Ours-FC does not achieve satisfying results. Recall that temporal feature maps in the proposed model are already multimodal representations since the sentence information has been integrated during the multimodal fusion process. Directly coupling the global sentence representation $\bar{s}$ with temporal feature units could possibly disrupt the visual correlations and temporal dependencies of the videos, which poses a negative effect on the temporal sentence grounding performance. In contrast, the proposed SCDM mechanism modulates the temporal feature maps by manipulating their scaling and shifting parameters under the sentence guidance, which is lightweight while meticulous, and still achieves the best results.

In addition, comparing Ours-SCM with Ours-SCDM, we can find that dynamically changing the modulation vectors $\gamma^c$ and $\beta^c$ with respect to different temporal feature units is beneficial, with R@5,IoU@0.7 increasing from 54.57% of Ours-SCM to 58.08% of Ours-SCDM. The SCDM intensifies meaningful words and cues in sentences catering for different temporal feature units, with the

Table 3: Ablation studies on the Charades-STA dataset (%).

| Method | R@1, IoU@0.5 | R@1, IoU@0.7 | R@5, IoU@0.5 | R@5, IoU@0.7 |
|---|---|---|---|---|
| Ours-w/o-SCDM | 47.52 | 26.91 | 69.85 | 49.35 |
| Ours-FC | 46.33 | 25.94 | 68.96 | 49.81 |
| Ours-MUL | 49.08 | 28.77 | 72.68 | 51.02 |
| Ours-SCM | 53.07 | 31.41 | 71.71 | 54.57 |
| **Ours-SCDM** | **54.44** | **33.43** | **74.43** | **58.08** |

motivation that different video segments may contain diverse visual contents and express different semantic meanings. Establishing the semantic interaction between these two modalities in a dynamic way can better align the semantics between sentence and diverse video contents, yielding more precise temporal boundary predictions.

## 4.6 Model Efficiency Comparison

Table 4 shows the run-time efficiency, model size (#param), and memory footprint of different methods. Specifically, "Run-Time" denotes the average time to localize one sentence in a given video. The methods with released codes are run with one Nvidia TITAN XP GPU. The experiments are run on the TACoS dataset since the videos in this dataset are relatively long (7 minutes on av-

Table 4: Comparison of model running efficiency, model size and memory footprint

| Method | Run-Time | Model Size | Memory Footprint |
|---|---|---|---|
| CTRL [10] | 2.23s | 22M | 725MB |
| ACRN [19] | 4.31s | 128M | 8537MB |
| Ours-SCDM | 0.78s | 15M | 4533MB |

erage), and are appropriate to evaluate the temporal grounding efficiency of different methods. It can be observed that ours-SCDM achieves the fastest run-time with the smallest model size. Both CTRL and ACRN methods need to sample candidate segments with various sliding windows in the videos first, and then match the input sentence with each of the segments individually. Such a two-stage architecture will inevitably influence the temporal sentence grounding efficiency, since the matching procedure through sliding window is quite time-consuming. In contrast, Ours-SCDM adopts a hierarchical convolution architecture, and naturally covers multi-scale video segments for grounding with multi-layer temporal feature maps. Thus, we only need to process the video in one pass of temporal convolution and then get the TSG results, and achieve higher efficiency. In addition, SCDM only needs to control the feature normalization parameters and is lightweight towards the overall convolution architecture. Therefore, ours-SCDM also has smaller model size.

## 4.7 Qualitative Results

Some qualitative examples of our model are illustrated in Figure 5. Evidently, our model can produce accurate segment boundaries for the TSG task. Moreover, we also visualize the attention weights (defined in Equation (2)) produced by SCDM when it processes different temporal units. It can be observed that different video contents attentively trigger different words in sentences so as to better align their semantics. For example, in the first example, the words "walking" and "open" obtain

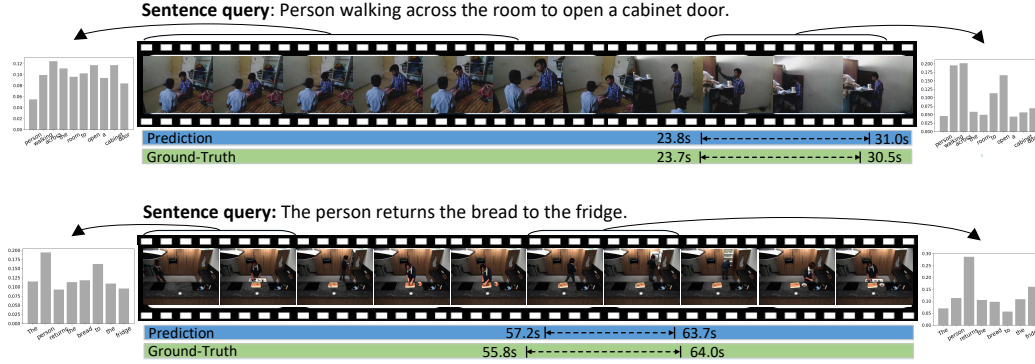

**Sentence query**: Person walking across the room to open a cabinet door.

Prediction               23.8s |------------------->| 31.0s
Ground-Truth        23.7s |------------------>| 30.5s

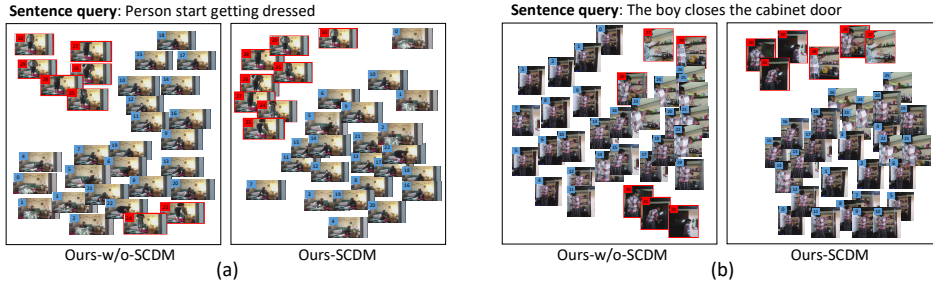

**Sentence query:** The person returns the bread to the fridge.

Prediction           57.2s |---------------->| 63.7s
Ground-Truth     55.8s |------------------->| 64.0s

Figure 5: Qualitative prediction examples of our proposed model. The rows with green background show the ground-truths for the given sentence queries, and the rows with blue background show the final location prediction results. The gray histograms show the word attention weights produced by SCDM at different temporal regions.

**Sentence query**: Person start getting dressed                 **Sentence query**: The boy closes the cabinet door

Ours-w/o-SCDM       Ours-SCDM              Ours-w/o-SCDM       Ours-SCDM

(a)                                         (b)

Figure 6: $t$-SNE projections of temporal feature maps yielded by the models Ours-w/o-SCDM and Ours-SCDM. Each temporal feature unit within these feature maps is represented by its corresponding video clip in the original video. Video clips marked with red color are within ground-truth video segments.

higher attention weights in the ground-truth segment since the described action indeed happens there. While in the other region, the word attention weights are more inclined to be an even distribution.

In order to gain more insights of our proposed SCDM mechanism, we visualize the temporal feature maps produced by the variant model Ours-w/o-SCDM and the full-model Ours-SCDM. For both of the trained models, we extract their temporal feature maps, and subsequently apply $t$-SNE [22] to each temporal feature unit within these maps. Since each temporal feature unit corresponds to one specific location in the original video, we then assign the corresponding video clips to the positions of these feature units in the $t$-SNE embedded space. As illustrated in Figure 6, temporal feature maps of two testing videos are visualized, where the video clips marked with red color denote the ground-truth segments of the given sentence queries. Interestingly, it can be observed that through SCDM processing, video clips within ground-truth segments are more tightly grouped together. In contrast, the clips without SCDM processing are separated in the learned feature space. This demonstrates that SCDM successfully associates the sentence-related video contents according to the sentence semantics, which is beneficial to the later temporal boundary predictions. More visualization results are provided in the supplemental material.

## 5 Conclusion

In this paper, we proposed a novel semantic conditioned dynamic modulation mechanism for tackling the TSG task. The proposed SCDM leverages the sentence semantics to modulate the temporal convolution operations to better correlate and compose the sentence-related video contents over time. As SCDM dynamically evolves with the diverse video contents of different temporal granularities in the temporal convolution architecture, the sentence described video contents are tightly correlated and composed, leading to more accurate temporal boundary predictions. The experimental results obtained on three widely-used datasets further demonstrate the superiority of the proposed SCDM on the TSG task.

# 6 Acknowledgement

This work was supported by National Natural Science Foundation of China Major Project No.U1611461 and Shenzhen Nanshan District Ling-Hang Team Grant under No.LHTD20170005.

## Footnotes

\*This work was done while Yitian Yuan was a Research Intern at Tencent AI Lab.

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
