[Supplementary Material]

# Supplemental Material for:
# Semantic Conditioned Dynamic Modulation
# for Temporal Sentence Grounding in Videos

**Yitian Yuan**[*]
Tsinghua-Berkeley Shenzhen Institute
Tsinghua University
yyt18@mails.tsinghua.edu.cn

**Lin Ma**
Tencent AI Lab
forest.linma@gmail.com

**Jingwen Wang**
Tencent AI Lab
jaywongjaywong@gmail.com

**Wei Liu**
Tencent AI Lab
wl2223@columbia.edu

**Wenwu Zhu**
Tsinghua University
wwzhu@tsinghua.edu.cn

This supplemental material includes the following contents:

- The ablation studies of the proposed model on the TACoS dataset.

- More qualitative results for temporal sentence grounding in videos.

- More visualization results of the $t$-SNE projections of the learned temporal feature maps.

## 1 Ablation Studies on the TACoS Dataset

Table 1 shows the ablation studies of our proposed model on the TACoS dataset. Overall, the results are in accord with the ablation studies on the Charades-STA dataset, which we have provided in the main paper. Our full model with semantic conditioned dynamic modulation achieves the best results, outperforming the Ours-SCM model which adopts the same modulation vectors over the whole feature map. It demonstrates that considering the video content diversity in the modulation procedure like SCDM is beneficial to temporal sentence grounding in videos. In addition, directly fusing the sentence representation into temporal feature maps like Ours-MUL and Ours-FC is also inferior to the semantic modulation. We can also find that Ours-MUL is better than Ours-FC. The reason is that Ours-MUL is actually a degraded form of the semantic conditioned modulation, with the sentence representation itself being $\gamma^c$ parameter. Without introducing sentence information to correlate and compose the sentence-related video contents in the temporal convolution procedure, Ours-w/o-SCDM still gets the worst results.

Table 1: Ablation studies on the TACoS dataset (%).

| Model | R@1,IoU@0.3 | R@1,IoU@0.5 | R@5,IoU@0.3 | R@5,IoU@0.5 |
|---|---|---|---|---|
| Ours-w/o-SCDM | 16.82 | 14.97 | 32.53 | 26.67 |
| Ours-FC | 18.42 | 14.94 | 35.72 | 28.54 |
| Ours-MUL | 19.26 | 15.81 | 36.01 | 29.15 |
| Ours-SCM | 24.16 | 20.72 | 38.61 | 31.17 |
| **Ours-SCDM** | **26.11** | **21.17** | **40.16** | **32.18** |

---

[*]This work was done while Yitian Yuan was a Research Intern at Tencent AI Lab.

## 2   More Qualitative Results

More qualitative results of our proposed model for temporal sentence grounding in videos can be found in Figure 1. Our proposed model can precisely locate the desired video segments corresponding to the given sentence queries, and the dynamic modulation procedure can meanwhile intensify some critical words or cues in sentences for grounding. For example, the objects "*knife*", "*cup & coffee*", and "*sandwich*" in (a), (b), and (c) examples, the action "*dismount*" in (d), and the person "*older man*" in (e) get higher attention weights in the target temporal regions. Those words are indeed important cues for identifying the target video segments. And intensifying their semantics in the SCDM procedure can better recognize and link the target-video contents over time and thus being beneficial to the subsequent temporal boundary predictions.

## 3   More Visualization Results of the Learned Temporal Feature Maps

More visualization results of temporal feature maps via $t$-SNE projection are provided in Figure 2. We can find that, after the proposed SCDM processing, the video clips within ground-truth video segments are more tightly grouped together in the learned feature space. It demonstrates that introducing sentence information to modulate the temporal convolution procedure can correlate the sentence-related video contents over time, and ease the following temporal boundary predictions performed over the feature maps.

Sentence query: The person gets out a knife.

Prediction 31.1s ◄------------► 41.2s
Ground-Truth 30.3s ◄------------► 41.5s

(a)

Sentence query: Person pouring coffee into a cup in the dining room.

Prediction 19.7s ◄-----------------------------► 30.6s
Ground-Truth 21.3s ◄-----------------------------► 30.7s

(b)

Sentence query: The person puts the sandwich into the refrigerator.

Prediction 7.5s ◄---------------------► 14.8s
Ground-Truth 7.9s ◄---------------------► 15.1s

(c)

Sentence query: He dismounts and raises his arms.

Prediction 88.3s ◄-------► 97.1s
Ground-Truth 88.2s ◄-------► 97.4s

(d)

Sentence query: An older man is helping him paint the fence.

Prediction 10.1s ◄-----------------------------------------------► 62.7s
Ground-Truth 11.4s ◄------------------------------------------------------► 70.2s

(e)

Figure 1: More qualitative results for temporal sentence grounding in videos.

Figure 2: $t$-SNE projections of temporal feature maps yielded by the models Ours-w/o-SCDM and Ours-SCDM. Each temporal feature unit within these feature maps is represented by its corresponding video clip in the original video. Video clips marked with red color are within ground-truth video segments. Zoom in to see details.