[Reviews · NeurIPS 2019]

Reviewer 1



the technical novelty of the paper is moderate. I can see some small differences between the proposed method and conditional BN or dynamic filters. my concern is that there is too much engineering for this task in terms of finding temporal dimensions, feature types and hyperparameters. However, empirically, the method shows promising results on three datasets. some related work are missing e.g. Gavrilyuk et al. Actor and Action Video Segmentation from a Sentence.

Reviewer 2



This paper proposes a semantic conditioned dynamic modulation (SCDM) mechanism for temporal sentence grounding in videos. The strengths are as follows: (1) Technically, the proposed SCDM is novel, and the idea of using sentence information to compose related video contents for temporal grounding is natural and reasonable. Coupling the SCDM mechanism in a hierarchical temporal convolutional network also makes the video contents with different temporal scales and granularities be linked together, and naturally supports the requirements of multi-scale video spans for temporal sentence grounding. Overall, this paper has good originality and quality from the technical aspect. More detail comments can be found in the summarized contributions above. (2) Extensive experiments on three public datasets show that the proposed SCDM significantly outperforms several state-of-the-art methods. The ablation studies and qualitative results are also convincing. (3) The proposed SCDM is light-weighted and concise, and the presentation of the paper is clear and this work is easy to follow. There are also some minor points: (1) Overall, the SCDM outperforms other baseline methods. But why the SCDM gets lower results on R@5, IoU@0.3 in the TACoS dataset, and lower results on R@5, IoU@0.5 in the Charades-STA dataset? Please provide more explanations. (2) In SCDM, the authors compute the word attention weights in the sentence. However, the word attention weights are not considered in the multi-modal fusion procedure. (3) What happens if the SCDM is only performed on several temporal convolutional layers, not on all the temporal convolutional layers? In addition, the authors are suggested to release the code for SCDM and facilitate further researches.

Reviewer 3



ORIGINALITY The originality of the work is somehow limited as it is porting the SSD approach for object detection to action localization. While I understand that the encoding of the sentence with the referring expression and the addition of the temporal dimension is novel, it is not a breakthrough idea either. (after author's feedback) The authors have better explained the novelty on how the sentence is encoded and linked with the video. The proposed approach is more novel than I understood after my original review. QUALITY The reported results are state of the art in the presented datasets. However, as there are no comparative results in terms of frame/second or size of the model or memory footprint, the results are somehow incomplete. CLARITY The text is a bit tedious to follow, especially when describing an architecture. SIGNIFICANCE The significance of the work seems limited beyond setting new state of the arts in three benchmarks.

[Author Response · NeurIPS 2019]

We sincerely thank all the three reviewers for their valuable comments, and what follow are our itemized responses.

**To Reviewer 1:**

**1. Regarding the differences between our SCDM and conditional BN or dynamic filter:** Conditional BN is exploited in style transfer and text-to-image synthesis, where the normalization vectors $\gamma$ and $\beta$ are shared across batch or instance. However, it is not easily amenable to the video grounding task. For grounding, the input sentences need to make detailed interactions with

Table 1: Performance comparison on Charades-STA (%).

| Method | R@1, IoU@0.5 | R@1, IoU@0.7 | R@5, IoU@0.5 | R@5, IoU@0.7 |
|--------|--------------|--------------|--------------|--------------|
| Ours-SCDM | 54.44 | 33.43 | 74.43 | 58.08 |
| Ours-DF | 45.63 | 25.45 | 70.47 | 48.52 |

different video temporal units and thereby determine accurate temporal boundaries. As such we proposed SCDM, in which the modulation parameters are explicitly generated, based on the sentence semantics, to manipulate temporal video features. The modulation procedure also dynamically evolves by attending to different words in sentences with respect to different temporal feature units, in order to establish detailed and accurate multimodal semantic interactions over time. Regarding dynamic filter, all the convolutional kernels are generated based on the inputs, which requires careful optimization tuning. Meanwhile, it also leads to larger model size and memory footprint. In contrast, our SCDM is more lightweight by controlling only the parameters of sentence-guided feature normalization. We also replace SCDM by dynamic filters, leading to inferior results, as shown in Table 1 above.

**2. Regarding the engineering part for this task:** For tackling the video grounding task, we follow previous work to determine the feature types. The setting of temporal dimensions follows the spirit of decaying them layer by layer (by half). For other hyper-parameters, we empirically set the filter number to 512, and also find that the performance is insensitive to this hyper-parameter. The trade-off hyper-parameters of the two loss terms are determined by balancing the numerical scales of them. As such, our model does not require too much tuning effort.

**3. Regarding missing related works:** Thanks for your reminding. We will include this work in our revised paper.

**4. Regarding feature types in Table 1:** In Table 1 of the main paper, CTRL, MCF, ACRN, ACL, TGN, and Xu et al. use C3D features, while SAD uses VGG16 features, and MAN uses I3D features. We adopt C3D features for the TACoS dataset following most methods. Since MAN achieves the best performance on the Charades-STA dataset, we also use I3D features on this dataset for fair comparisons. We will clarify the feature types in our revised paper.

**To Reviewer 2:**

**1. Regarding SCDM results under different IoUs:** The lower results for low IoUs on Charades-STA and TACoS are mainly due to the biased annotations. For example, in Charades-STA, the annotated ground-truth segments are 10s on average while the video duration is only 30s on average. Randomly selecting one candidate segment can also achieve high R@5, IoU@0.5 value as 0.5435 (even comparable with CTRL), but the R@5,IoU@0.7 is much lower as 0.2065. It indicates that the Recall values under higher IoUs are more stable and convincing even considering the dataset biases.

**2. Regarding computing word attention in the multimodal fusion:** We also tried computing word attention in the multimodal fusion but found no improvements. Since the multimodal fusion aims to let each video clip meet and interact with the general sentence semantics and does not directly serve for temporal boundary predictions, using globally averaged word features seems to be enough for this stage.

**3. Regarding SCDM only performed on several temporal convolutional layers:** The performance degenerates if SCDM is only performed on several temporal convolutional layers. Since each temporal convolutional layer corresponds to one specific temporal scale, discarding SCDM in any layer will compromise the prediction accuracy of that scale.

**To Reviewer 3:**

**1. Regarding missing related works:** Thanks for your reminding. We will include this work in our revised paper.

**2. Regarding run-time, model size and memory footprint:** Table 2 shows the average run-time to localize one sentence, model size (#param) and memory footprint. The methods with released codes are compared on one Tesla M40 GPU. It can be observed that ours-SCDM achieves the fastest run-time with the smallest model size. We will release the code if accepted.

Table 2: Comparison of model running efficiency and model size.

| Method | Run-Time | Model Size | Memory Footprint |
|--------|----------|------------|------------------|
| CTRL | 3.75s | 22M | 1214MB |
| ACRN | 5.29s | 128M | 8432MB |
| ACL | 4.52s | 23M | 1458MB |
| Ours-SCDM | 0.81s | 15M | 4481MB |

**3. Regarding original contributions of the proposed approach:** Temporal sentence grounding in video differs from the traditional action detection in the sense that the former provides an explicit sentence guidance for determining target video segments. Therefore, how to fully establish the semantic interactions between video and sentence, and fully leverage the sentence semantics to detect and link corresponding video contents over time are very crucial. To solve these issues, our model is not one trivial extension of SSD. The proposed SCDM leverages the sentence information to control the modulation parameters of the feature normalization procedure in the hierarchical convolution architecture, instead of simply fusing the sentence features and video features. Such a sentence guided temporal feature modulation stimulates the temporal convolution operation to link sentence-related video contents over time. The modulation of temporal features also dynamically evolves for different video contents, enabling better multimodal alignment over time to support more precise boundary predictions. Moreover, our proposed SCDM is lightweight, and achieves the superior performance compared against the state-of-the-art approaches.

[Meta-Review · NeurIPS 2019]

This paper proposes a mechanism for conditioning temporal convolutions on a sentence embedding in the context of aligning sentences with video segments. The reviewers agree that this is solid work with good experimental results. The novelty of the work appears limited to the context of the sentence grounding task, and as such is somewhat incremental. However the reviewers highlight the efficiency of the approach in terms of memory and computation, and feel the results will be of interest to vision and language researchers.